# Joining Strength of Self-Piercing Riveted Vibration-Damping Steel and Dissimilar Materials

Keong Hwan Cho [1,2], Jin Hyeok Joo [1,2], Min Gyu Kim [1,3], Dong Hyuck Kam [1,*] and Jedo Kim [4,*]

1   Advanced Welding & Joining R&BD Group, Korea Institute of Industrial Technology,
    Incheon 21999, Republic of Korea
2   Department of Mechanical Design Engineering, Hanyang University, Seoul 04763, Republic of Korea
3   Department of Mechanical Convergence Engineering, Hanyang University, Seoul 04763, Republic of Korea
4   Department of Mechanical and Systems Design Engineering, Hongik University,
    Seoul 04066, Republic of Korea
*   Correspondence: kamdong@kitech.re.kr (D.H.K.); jedokim@hongik.ac.kr (J.K.)

**Abstract:** A vibration-damping steel panel is used for lightweight vehicles to block any noise subjected to the passenger cabin replacing heavy fiber-based insulators. Conventional weld joining methods often encounter problems due to the presence of viscoelastic compounds reducing the joint quality and making the joining process unproductive. In this work, we present experimental results that show the self-piercing riveting (SPR) process can be used to produce high-quality joints between vibration-damping steel and (i) commonly used steel alloy (SPFC590DP), (ii) carbon-fiber-reinforced-plastic (CFRP) panels. Various die shapes are used to investigate the resulting interlock width and bottom thickness of the joints and tensile shear load tests were performed to evaluate the joining strength. The results show that high-quality joints between vibration-damping steel and the steel alloy are possible for all the dye types and panel configurations, used in this study, producing up to 6.2 kN of tensile shear load. High-quality joints were also possible with CFRP producing up to 4.0 kN, however, acceptable joints were formed only when the CFRP panels were on top during the riveting process due to severe cracking.

**Keywords:** self-piercing riveting (SPR); dissimilar joining; vibration-damping steel; carbon-fiber-reinforced plastic (CFRP); joint quality





## 1. Introduction

Vibration-damping steel is used for various industrial applications used to reduce noise and vibration through the visco-elastic sandwiched viscoelastic adhesive [1,2]. Nowadays, in response to environmental regulations, its use is widening to many areas of automobiles comprising cowl and dash panels replacing conventional heavy cloth-based insulation. Many lightweight materials have already been applied to meet the ever-increasing regulations such as advanced high-strength steel, aluminum alloy, and composite [3,4], and when new materials are added to the existing framework, a new joining method is necessary to establish sufficient strength between the material and production rates. Consequently, the dissimilar material joining of the vibration-damping steel to various other materials is becoming one of the key issues to realize versatile designs for next-generation of environmentally friendly mobile vehicles. In particular, joints with conventional steel panels and increasingly adopted carbon-fiber-reinforced plastic (CFRP) is of interest, and, accordingly, it is crucial to develop reliable dissimilar joining methods of these materials to the vibration-damping steel panels.

The self-piercing riveting (SPR) process is completed by pushing semi-tubular rivets using a punch drilling through the top panel of the joint while subsequently creating an interlock between the rivet shank and bottom panel [5–7]. The SPR process has been already widely adopted for joining dissimilar materials such as steel–aluminum lap joints

because it is simple and produces high-quality joints [8–14]. Studies have found that the joint strengths of SPR joints are comparable or superior to that of conventional joining methods such as resistance spot-welding (RSW) [6] and laser welding techniques which are frequently used in the automotive industry for combinations of Al/Fe/Ti/Cu [15]. Although there have been recent successes in laser welding of CFRP with steel [16], multi-material integration of high-strength and brittle materials that melt cannot be formed is in need for manufacturing new vehicles [17]. When the material to be joined becomes brittle, the base material fractures, rather than plastically deform making SPR joining process increasingly difficult. To investigate the rivetablity of various materials and to optimize the joining strength of SPR rivets, many previous studies have been conducted. Ma et al. [11] reported the effect of the rivet and die on self-piercing riveted joints for aluminum alloy (AA6061) and mild steel (CR 4). The self-piercing riveted joints between aluminum alloy (A5052-H34) and mild steel (SPCC) were studied by Abe et al. experimentally and numerically [12]. The study found that the steel panel on top/aluminum panel on bottom configuration resulted in higher quality joint characteristics compared to the reverse configuration. Han et al. [13] found that the panel configuration during the riveting process has a substantial impact on the mechanical characteristic of the formed joints between multi-layered aluminum (AA6111 and NG5754) and steel alloy (HSLA350) panels. Wood et al. [14] investigated tension, shear, and peel characteristics of the self-piercing riveted joints for aluminum sheets (A5754) and found that interlock failure was the main issue, and the shear performance was significantly degraded at high-speed car crash test conditions. A comprehensive experimental and numerical investigation of the role of the rivet and the die design was completed by Karathanasopoulos et al. on aluminum/steel sheets [18]. A recent study by Huang et al. investigated the effect of shot peening on static and fatigue properties of SPR joints and found that increased joint strength is expected compared to that of non-shot peened SPR joints [19]. Wang et al. showed an SPR riveting method for carbon-reinforced plastic (CFRP) and aluminum ally sheets and found post-curing SPR method resulted in superior joint quality compared to conventional SPR methods [20]. Our group recently presented experimental results showing conditions in which good SPR joint quality between vibration-damping steel/aluminum, and vibration-damping aluminum/steel/carbon-reinforced plastic (CFRP) [21,22]. However, to the best of the author's knowledge, no study has been reported for the self-piercing riveted joint formed between vibration-damping steel and commonly used SPFC590DP steel alloy panels/CFRP. The viscoelastic adhesive in between the vibration-damping steel makes the dissimilar joining difficult since the adhesive layer introduces inhomogeneities during many joining methods including the SPR process [21].

In this study, we investigate the quality of the self-piercing riveted dissimilar joints between the vibration-damping steel and SPFC590DP/CFRP. The effect of the die types and panel placement configuration on the geometrical indexes parameters such as interlock width and bottom thickness is evaluated for each self-piercing riveted joint along with the mechanical performance characteristics.

## 2. Materials and Methods

The materials used for the experiment presented in this paper are vibration-damping steel, steel alloy (SPFC590DP or 590DP), and thermoset carbon-fiber-reinforced plastic (CFRP) (7-cross-ply (0°/90°) laminated structure in an epoxy matrix). The chemical composition of the steel alloy is shown in Table 1 and the composition suggests it is a widely used steel panel in the automotive industry. As shown in Figure 1, the vibration-damping steel consists of two 0.7 mm thickness SFC590DP sheets with a 0.1 mm thickness viscoelastic adhesive polymer layer in between resulting in a total thickness of 1.5 mm. The tensile strength and thickness of the materials are listed in Table 2. The tensile strength of all materials is evaluated under the ASTM E8M specification.

**Table 1.** Chemical composition of the base steel alloy used in this study.

| Material | Chemical Composition (wt.%) | | | | | |
|---|---|---|---|---|---|---|
| | C | Si | Mn | P | S | Fe |
| SPFC590DP | 0.07 | 0.14 | 1.44 | 0.013 | 0.002 | 98.335 |

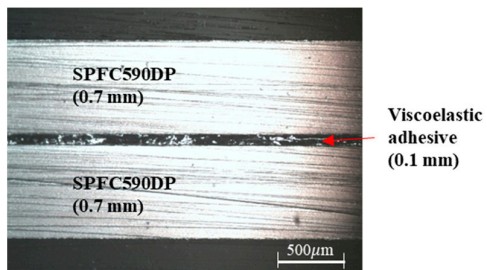

**Figure 1.** The cross-sectional view of the vibration-damping steel panel.

**Table 2.** The tensile strength and the thickness of the materials used in this study.

| Sheet Material | Ultimate Tensile Strength (MPa) | Thickness (mm) |
|---|---|---|
| Vibration-damping steel | 618 | 1.5 |
| SPFC590DP | 609 | 1.4 |
| CFRP | 1032 (0°)/234 (45°)/1014 (90°) | 1.8 |

Self-piercing riveted joints in this study are formed with a hydraulic-type riveting machine (Rivset Gen2, BÖLLHOFF, Bielefeld, Germany) that has a maximum setting force of 78 kN. The rivet (boron steel with Almac®, Craigavon, Northern Ireland, UK, coating (a combination of Zn, Sn, and Al), 480 ± 30 HV in hardness, supplied by BÖLLHOFF) with flat countersunk was used. The shape and dimensional information of the rivet and the geometrical information of the dies are shown in Figure 2. A rivet shank diameter and rivet length are 5.3 mm and 5.0 mm, respectively. The rivet length of 5.0 mm is selected to be approximately 2 mm thicker than the total thickness of the base materials (2.9 or 3.3 mm). Four die types (also supplied by BÖLLHOFF) are tested to investigate the effect of the die geometry on the joining performance. Type A1 (diameter: 8.8 mm, depth: 1.8 mm) and type A2 (diameter: 9.2 mm, depth: 1.8 mm) dies have a basic flat-bottom shape. In addition, both type B (diameter: 9.5 mm, depth: 1.8 mm) and type C (diameter: 9.0 mm, depth: 2.0 mm) have intrusions at the center of the die cavity. The intrusion is a conical shape for type B and the height is smaller than the depth of the die cavity. The intrusion height is larger than the depth of the die cavity for die type C, resulting in more extrusion and we will designate this shape as the nipple.

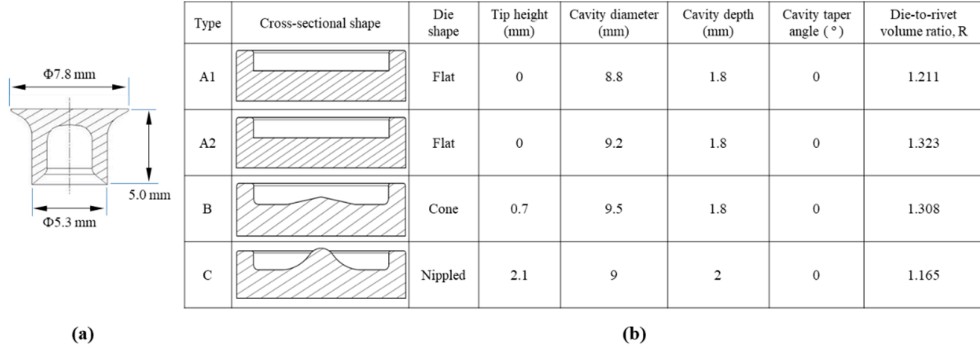

(a)          (b)

**Figure 2.** (**a**) The geometric dimensional parameters of the SPR rivet and (**b**) the die types used in this study.

Cross-sectional analyses of the self-piercing riveted joint were carried out to determine the geometrical features of the joints which is one of the important inspections that determine the acceptability of the formed joints [7]. Two geometrical indexes, shown in Figure 3, are measured: (1) interlock width, $w_I$ (the distance from the tip of the deformed rivet shank to the pierced point of the top panel), (2) bottom thickness, $t_B$ (remaining thickness of the bottom panel after the riveting process is complete). The setting forces are set between 35–57 kN for vibration-damping steel and SPFC590DP self-piercing riveted joints and 53–60 kN for vibration-damping steel and CFRP. All the measurements were taken using an optical microscope system with an image analysis tool.

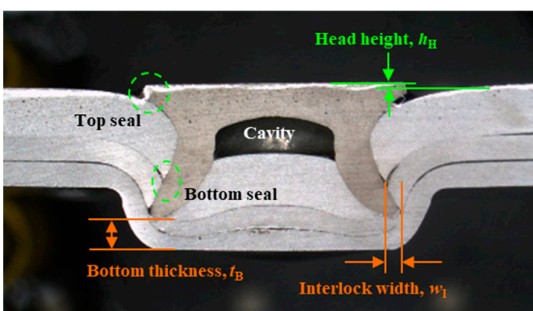

**Figure 3.** The definition of geometrical indexes of an SPR joint.

The joint strength of the SPR joint is examined by tensile shear test (test machine: Shimadzu Universal Testing Machine AG-300KNX, maximum load = 30 tons). The test procedure and the sample dimensions for the test conformed to KS B ISO 14273 standards and the test sample is shown in Figure 4. Crosshead speed of 5 mm/min was used and the average value of three samples are presented here but the consistency is maintained for tests up to 10 samples.

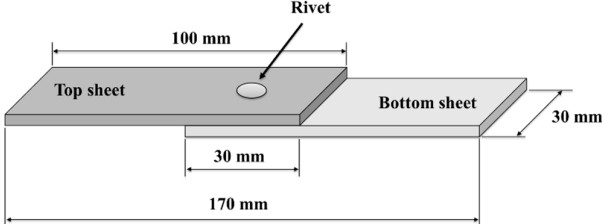

**Figure 4.** The schematic diagram of the tensile shear test specimen (KS B ISO 14273).

### 3. Results and Discussion

*3.1. Self-Piercing Riveted Joint of Vibration-Damping Steel and SPFC590DP*

Figure 5 shows the cross-section of the joints formed using the various types of die geometries shown in Figure 2 for two different panel configurations, namely, vibration-damping steel (top)/SPFC590DP (bottom) and SPFC590DP (top)/vibration-damping steel (bottom). Good interlock between the two sheets is observed for all die types and panel configurations. The figure shows that when the vibration-damping steel panel is on top, there exists some separation between the vibration-damping panel sheets within the rivet. The figure also shows that when the SPFC590DP panel is on top, the cavity formed within the rivet after joining can be significantly large for flat die types. Such separations and cavities can be mitigated using different die geometries such as the nipple type. Nevertheless, the subsequent tensile shear test reveals that any separations and cavities do not affect the overall joint quality. The cross-section images reveal that there are minimal microstructural changes in both the rivet and the base material. This is expected since no external heat is applied to the SPR process, and the extent of plastic deformation is below the fracture point of the base material. Any microstructural change should be

avoided in the SPR process since inhomogeneous material properties are not desirable to ensure joint quality consistency.

| Die Type | A1 (Flat) | A2 (Flat) | B (Cone) | C (Nippled) |
|---|---|---|---|---|
| Vib. (Top) / SPFC590DP (Bottom) | (a) | (b) | (c) | (d) |
| SPFC590DP (Top) / Vib. (Bottom) | (e) | (f) | (g) | (h) |

**Figure 5.** The cross-sectional images of the self-piercing riveted joint between vibration-damping steel and SPFC590DP with respect to die types and panel configurations: (**a**,**e**) die type A1, (**b**,**f**) die type A2, (**c**,**g**) die type B, (**d**,**h**) die type C.

Figure 6a,b show the interlock width ($w_I$), bottom thickness ($t_B$), and tensile shear loads for the joints formed using the SPR process between vibration-damping steel and SPFC590DP under different die types. The results show that the joints formed using die type B do not meet the minimum requirement for $w_I \geq 0.2$ mm (interlock width ($w_I$) $\geq 0.2$ mm, bottom thickness ($t_B$) $\geq 0.2$ mm [7]). $w_I$ was found to be shorter for die type B compared to other types while $t_B$ was found to be shorter for die type C under vibration-damping steel panel on the bottom configuration. The tensile shear load test results shown in the figure shows that all die types and panel configurations show good joint formation producing between 5.0 to 6.0 kN of tensile shear load which is excellent for producing bodies for motor vehicles compared with SPR joints for other materials [23–29].

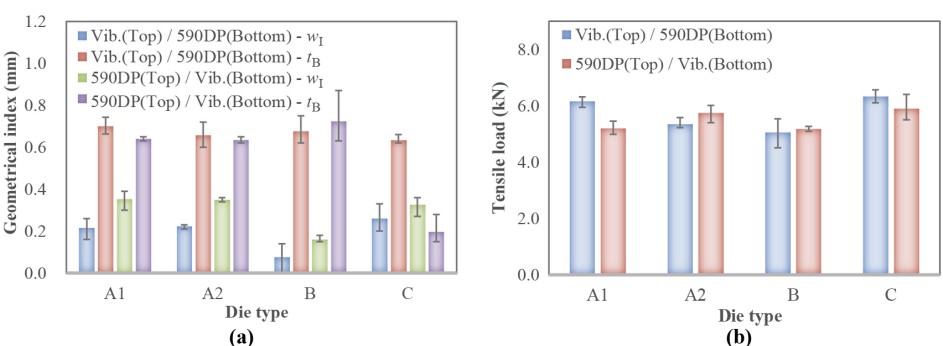

**Figure 6.** (**a**) $w_I$ and $t_B$ and (**b**) the tensile shear loads for self-piercing riveted joint between vibration-damping steel and SPFC590DP or 590DP with respect to the die types.

The failure modes of the specimen after the tensile shear load test were investigated and the results are shown in Figure 7 for all the different types of die and panel configurations. When the vibration-damping steel panels are on top, all the joint failure was due to rivet pull out which can be seen in the photographs. In contrast, when the vibration-damping steel panels are on the bottom, the failure was due to bottom sheet failure except for the die type A1 which was due to interfacial delamination failure of the bottom vibration-damping steel sheet. The failure modes show consistency among the die types studied in this study and the tensile shear load test results, shown in Figure 6, indicate that there is not a significant joint quality difference between the two-panel configurations.

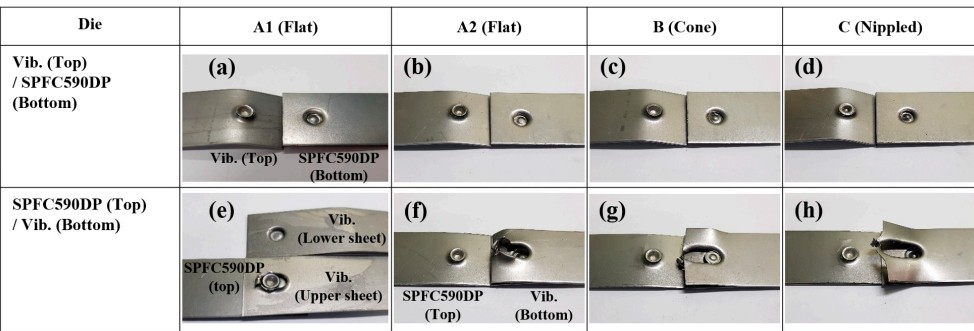

**Figure 7.** The failure modes of self-piercing riveted joints formed between vibration-damping steel and SPFC590DP with respect to the die types and the panel configurations. (**a**,**e**) die type A1, (**b**,**f**) die type A2, (**c**,**g**) die type B, (**d**,**h**) die type C.

To investigate any geometrical irregularities for self-piercing riveted joints formed between vibration-damping steel and regular SPFC590DP panels, the cross-sectional views of the three configurations (vibration-damping steel (top)/SPFC590DP (bottom), SPFC590DP (top)/vibration-damping steel (bottom), and SPFC590 (top)/SPFC590 (bottom), were compared. The results in Figure 8 show that for all three configurations, good bottom panel penetration was observed indicating excellent joint formations. No irregular panel deformation was observed besides the relatively large cavities formed within the rivet for the panel configuration when the SPFC590DP panel is on top. The large cavities are found in other SPR-joined high-strength materials since the brittleness of the material resists deformation during the riveting process [22]. We found that as long as the top seal is intact, the cavities within do not affect the joint quality and the long-term fatigue characteristics of the joints.

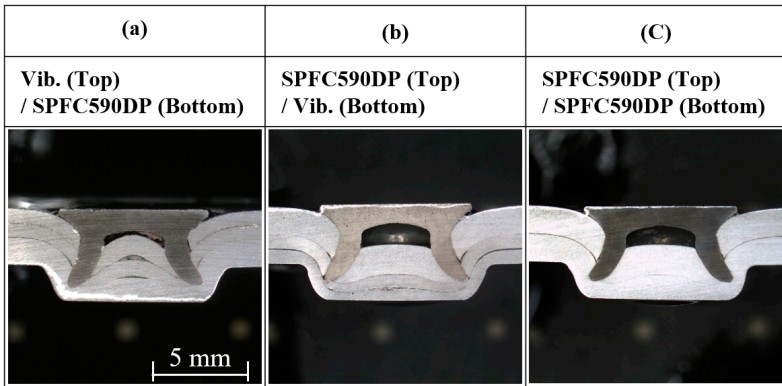

**Figure 8.** Comparison of the cross-sectional images of self-piercing riveted joints (fabricated by type A1 die) for different panel stacking combinations. (**a**) Vibration-damping steel (top) and SPFC590DP (bottom), (**b**) SPFC590DP (top) and vibration-damping steel (bottom), and (**c**) SPFC590DP (top) and SPFC590DP (bottom).

The geometrical index values and the tensile shear load test results are shown in Figure 9a,b, respectively. The results show that adequate interlock width and bottom thickness are observed for all three configurations indicating good joint formation. The tensile shear test results also show that, on average, values between 5.5 to 7.5 kN are exhibited indicating excellent joint quality. The failure modes show rivet pullout for all three configurations, however, when the vibration-damping steel panel is on the bottom, only the bottom sheet is separated as seen in the middle photograph.

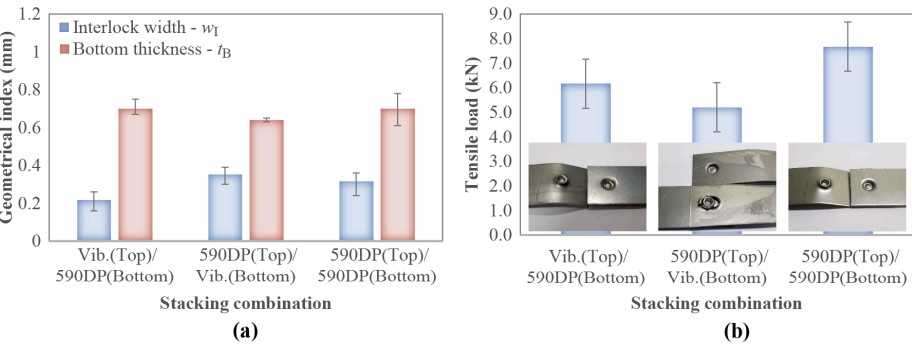

**Figure 9.** (**a**) $w_I$ and the $t_B$ and (**b**) the tensile shear loads for self-piercing riveted joint between vibration-damping steel and steel alloy under different panel stacking combinations.

### 3.2. SPR Joint of Vibration-Damping Steel and CFRP

Figure 10 shows the cross-section of the joints formed using various die types shown in Figure 2 for CFRP on top and vibration-damping steel panel on the bottom. The CFRP on the bottom case is not presented here because, due to the brittle nature of CFRP, severe fracture resulted in the bottom panel after the riveting process resulting in under par joint formations. We find that for all die types, severe cracking and tearing are shown on the bottom CFRP panel. The figure shows that within the cavity of the rivet, the CFRP is not plastically deformed as seen from steel alloys in Figure 8 but rather it is cracked and sandwiched between itself. Such brittleness nature of CFRP does not allow for it to make a good interlock when the panel is on the bottom. Nevertheless, the results for the CFRP panel on the top show that for all die types, a good interlock with the bottom vibration-damping panel can be achieved. The photographs show that there is significant cracked CFRP within the rivet, however, the subsequent tensile shear load test result indicates that its effect on the joint quality is marginal, and as long as the top seal is secure, we do not expect any long-term detrimental defect on the fatigue strength of the joints.

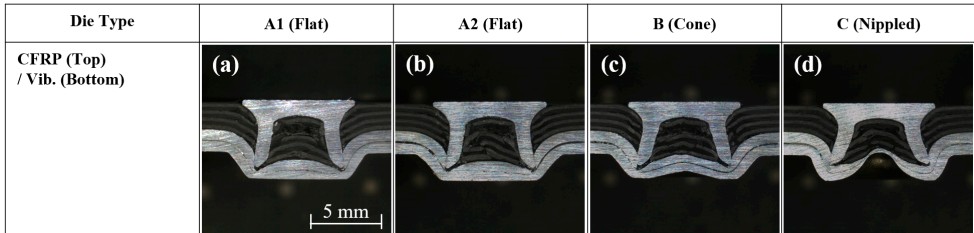

**Figure 10.** Cross-sectional images for the self-piercing riveted joint formed between vibration-damping steel and CFRP with respect to the die types: (**a**) die type A1, (**b**) die type A2, (**c**) die type B, and (**d**) die type C.

The geometrical index parameters and tensile test results are shown in Figure 11a,b, respectively. The results show that inconsistent interlock width and bottom thicknesses result for the different die types varying up to 400%. The brittle nature of the CFRP results in significant differences in the joint geometries, especially for the interlock width. Only the joints formed using die type A1 meet the requirement for both $w_I$ and $t_B$. Yet, the tensile shear load test results for joints formed using die type C show the highest tensile shear load of 3.9 kN which is comparable to SPR-processed joints for CFRP [23,30]. Such a good result can be the result of the die-to-volume ratio, $R$, where the value for die type C is 1.165, the smallest of the die types used in this study. The smaller $R$ means that the rivet can fit tightly within the die during the interlock deformation resulting in a stronger joint formation.

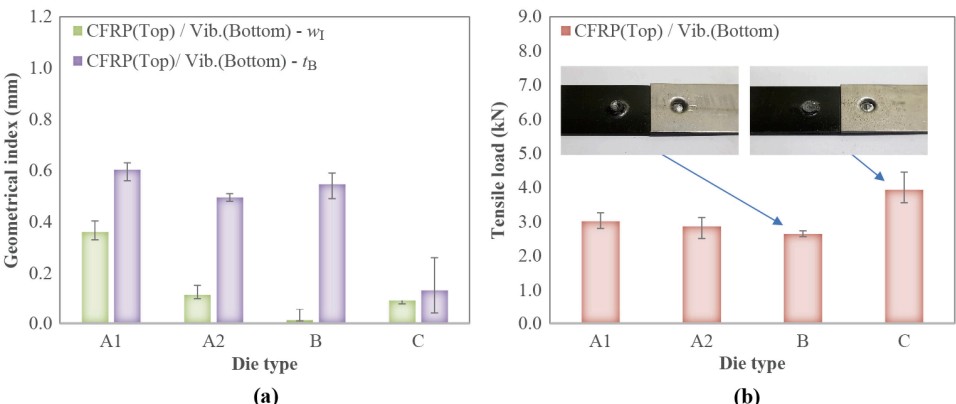

**Figure 11.** (**a**) $w_I$ and $t_B$ and (**b**) tensile shear loads for self-piercing riveted joints between CFRP and vibration-damping steel panels under different die types.

The comparison of the cross-sectional views of self-piercing riveted joints for two different panel configurations (CFRP on top/vibration-damping steel on the bottom and CFRP on top/SPFC590DP on the bottom) is shown in Figure 12. The photographs show that a good interlock between both vibration-damping steel and regular high-strength steel panel is possible with the SPR technique. However, due to the high punch force required to penetrate the high-strength CFRP, Figure 12b shows significant bottom panel cracking. Again, the brittle nature of high-strength steel is responsible for the cracking, but the subsequent analysis of the tensile shear load test reveals that its effect on the joint strength is negligible. Nonetheless, due to such cracking of the panel, the SPR process is not suitable for CFRP/SPFC590DP since the cracks allow for a higher probability of corrosion that significantly reduces their long-term corrosion resistance [7,31].

| (a) | (b) |
|---|---|
| **CFRP (Top) / Vib. (Bottom)** | **CFRP (Top) / SPFC590DP (Bottom)** |

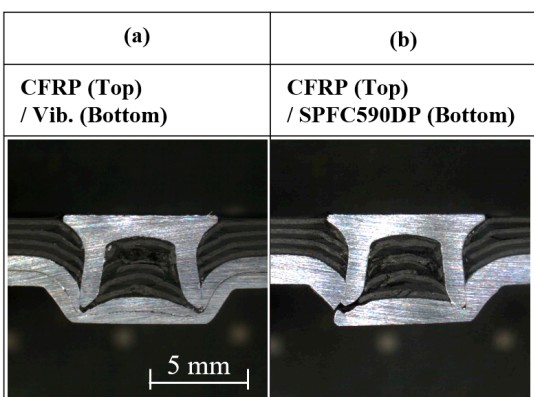

**Figure 12.** Comparison of the cross-sectional images of self-piercing riveted joint (fabricated by type A1 die) for different panel stacking configurations. (**a**) CFRP (top) and vibration-damping steel (bottom) and (**b**) CFRP (top) and SPFC590DP (bottom).

The geometrical index parameters and the tensile shear load test for the joints formed between CFRP and the two different kinds of steel panels using the SPR technique are shown in Figure 13. After several iterations, geometrical indexes were measured to be significantly lower for the case when the SPFC590DP panel is on the bottom when joined. The bottom thickness showed the largest deviance between the two configurations since the cross-section of the case when the SPFC590DP panel on the bottom often showed cracks as shown in Figure 12b. The tensile shear load tests suggest that there is a negligible difference in the joint strength between the two configurations. These results also confirm that when there is an adequate interlock between the two panels that are to be joined, the geometrical index parameters of the form joint have a marginal effect on the joint strength.

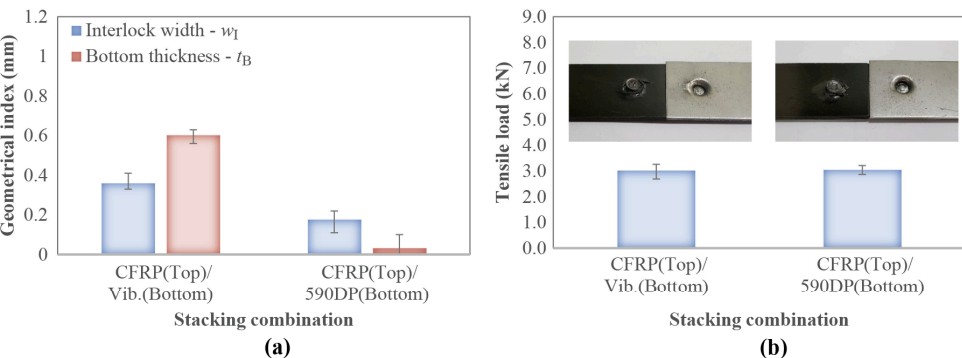

**Figure 13.** (**a**) $w_I$ and $t_B$ and (**b**) tensile shear loads for self-piercing riveted joints between vibration-damping steel and CFRP under different panel stacking configurations.

## 4. Conclusions

The self-piercing riveted joints between vibration-damping steel panels with commonly used SPFC590DP steel panels and carbon fiber reinforced plastic (CFRP) are studied in this work to evaluate the feasibly of using the SPR process on multi-material integration, especially for state-of-the-art mobility applications. From this study the following conclusions can be made:

1. Successful interlock between vibration-damping steel and regular SPFC590DP panels is possible using the SPR process free of any cracking under various die types (flat, coned, nippled).
2. The measured geometrical index parameters are within acceptable values, resulting in consistent tensile shear loads between approximately 5.0–6.2 kN. Such values were slightly lower than the joints formed between two SPFC590DP panels which were measured to be approximately 7.5 kN but still acceptable values for use in the automotive industry.
3. Successful interlocks between CFRP and the vibration-damping steel panels were possible only when the CFRP panel was positioned on top. Acceptable geometrical index parameters were measured resulting in tensile shear loads between 3.0–4.0 kN depending on the die type used.
4. Bottom cracking was observed for joints between CFRP and SPFC590DP panels which is not acceptable for commercial use of SPR process joining. Nevertheless, the geometrical index parameters and the tensile shear load test results were within acceptable values.

**Author Contributions:** Conceptualization, D.H.K. and J.K.; methodology, D.H.K.; formal analysis, K.H.C. and D.H.K.; investigation, K.H.C., J.H.J. and M.G.K.; resources, D.H.K.; data curation, J.K.; writing—original draft preparation, J.K.; writing—review and editing, J.K.; visualization, J.H.J. and M.G.K.; supervision, D.H.K. All authors have read and agreed to the published version of the manuscript.

**Funding:** The authors thank the following agencies for their support: the Republic of Korea Evaluation Institute of Industrial Technology and the Industry Core Technology Development Program funded by the Ministry of Trade, Industry and Energy (Korea) (KITECH EO-23-0007, 2015R1A6A1A03031833).

**Data Availability Statement:** Not applicable.

**Conflicts of Interest:** The authors declare no conflict of interest.

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
