# Peer review of "Joining Strength of Self-Piercing Riveted Vibration-Damping Steel and Dissimilar Materials"

_jmmp, doi:10.3390/jmmp7020065_

Round 1
Reviewer 1 Report
Thank you for this interesting work on searching for alternative solutions to welding processes or adhesives to join dissimilar materials. Nevertheless, in my opinion, this paper needs two significant improvements:
· Since claims are made about other joining processes, a comparison should be made between the “joint” strength obtained with the self-piercing rivets and other conventional processes/methods.
· More importantly, to be able to compare it, please quantify the stress. Otherwise, just by comparing the shear force, it is impossible to compare it directly with other means of joining. Therefore, it is necessary to account for the “joint” area.
Author Response
Dear reviewer, we have made the following improvements per your comments. Also, additional improvements were made and are highlighted in the new version of the manuscript.
- Since claims are made about other joining processes, a comparison should be made between the “joint” strength obtained with the self-piercing rivets and other conventional processes/methods.
We thank the reviewer for the suggestion. The last paragraph of page 1 (first paragraph of page 2) now includes the following sentence to acknowledge the joint strength compared to other conventional joining methods such as resistance spot-welding(RSW) “Studies have found that the joint strengths of SPR joints are comparable or superior to that of conventional joining methods such as resistance spot-welding (RSW)[6].” Please note that many conventional joining methods cannot be used to join dissimilar materials and only the references for the same materials (Al-Al or Steel-Steel) are found. Therefore, we feel that the SPR study presented in this work is of importance to the state-of-the-art lightweight vehicle industry.
- More importantly, to be able to compare it, please quantify the stress. Otherwise, just by comparing the shear force, it is impossible to compare it directly with other means of joining. Therefore, it is necessary to account for the “joint” area.
Thank you for pointing out that stress is important in determining the joint strength. Our experimental study conforms to the ASTM E8M specification so that the sample sizes are standardized. Therefore, in this field, it is customary to only look at the shear force. We would like to directly compare the joint strength of other conventional methods, however, for the dissimilar material combinations studied in this study, other methods are incapable of maintaining acceptable joints (form-wise). Thus dissimilar material joining is mostly done by SPR process and this field of study is relatively new due to the development of new materials such as vibration-damping steel, CFRP, etc.
The introduction includes the following paragraph:
“ The self-piercing riveting (SPR) process is done by pushing semi-tubular rivets using a punch drilling through the top panel of the joint while subsequently creating an interlock between the rivet shank and bottom panel [5-7]. The SPR process has been already widely adopted for joining dissimilar materials such as steel-aluminum lap joints because it is simple and produces high-quality joints [8-14]. Studies have found that the joint strengths of SPR joints are comparable or superior to that of conventional joining methods such as resistance spot-welding (RSW)[6]. Yet, the recent need for multi-material integration poses a new challenge since high-strength and brittle materials are increasingly being used to manufacture new vehicles [15]. When the material to be joined becomes brittle, the base material fractures, rather than plastically deform making SPR joining process increasingly difficult. To investigate the rivetablity of various materials and to optimize the joining strength of SPR rivets, many previous studies have been conducted. Ma et al. [11] reported the effect of the rivet and die on self-piercing riveted joints for aluminum alloy (AA6061) and mild steel (CR 4). The self-piercing riveted joints between aluminum alloy (A5052-H34) and mild steel (SPCC) were studied by Abe et al experimentally and numerically [12]. The study found that the steel panel on top/aluminum panel on bottom configuration resulted in higher quality joint characteristics compared to the reverse configuration. Han et al. [13] found that the panel configuration during the riveting process has a substantial impact on the mechanical characteristic of the formed joints between multi-layered aluminum (AA6111 and NG5754) and steel alloy (HSLA350) panels. Wood et al. [14] investigated tension, shear, and peel characteristics of the self-piercing riveted joints for aluminum sheets (A5754) and found that interlock failure was the main issue, and the shear performance was significantly degraded at high-speed car crash test conditions. A comprehensive experimental and numerical investigation of the role of the rivet and the die design was done by Karathanasopoulos et al. on aluminum/steel sheets[16]. A recent study by Huang et al. investigated the effect of shot peening on static and fatigue properties of SPR joints and found that increased joint strength is expected compared to that of non-shot peened SPR joints [17]. Wang et al. showed a SPR riveting method for carbon-reinforced plastic (CFRP) and aluminum ally sheets and found post-curing SPR method resulted in superior joint quality compared to conventional SPR methods [18]. Our group recently presented experimental results showing conditions in which good SPR joint quality between vibration-damping steel/Aluminum, and vibration-damping aluminum/steel/carbon-reinforced plastic (CFRP) [19,20]. However, to the best of the author’s knowledge, no study has been reported for the self-piercing riveted joint formed between vibration-damping steel and commonly used SPFC590DP steel alloy panels/CFRP. The viscoelastic adhesive in between the vibration-damping steel makes the dissimilar joining difficult since the adhesive layer introduces inhomogeneities during many joining methods including the SPR process [19].”

Reviewer 2 Report
In this work, the authors use the self-piercing riveting (SPR) process to join vibration-damping steel and SPFC590DP/CFRP) panels. The authors claim that the SPR can solve the problems encountered by conventional weld joining methods due to the presence of viscoelastic compounds reducing the joint quality and making the joining process unproductive. the influence of die shapes on interlock width, bottom thickness of the joints and tensile shear load tests were systematically studied, but in my opinion, these are all macro studies, not related to welding metallurgy. Due to the lack of interfacial metallurgical bonding and microstructure evolution studies, it is not possible to indicate whether the formation of compounds is reduced based on the available data in the work. Thus, I can not recommend it to publish at current form.
(1) It is recommended to include composition of the steels.
(2) The presentation of the images is well.
(3) The interfacial metallurgical bonding and microstructure evolution of the joints should be addressed.
(4) What is the shear stability of the joint? How many shear tests were done? This can be explained in the Materials and Methods.
Author Response
Reviewer 2
Dear reviewer, we have made the following improvements per your comments. Also, additional improvements were made and are highlighted in the new version of the manuscript.
1) It is recommended to include composition of the steels.
The first paragraph of materials and method section now includes the following sentence: “The chemical composition of the steel alloy is shown in Table 1.” and a new table is includes as Table 1.
Table 1. Chemical composition of the base steel alloy used in this study.
|
Material |
Chemical composition (wt. %) |
|||||
|
SPFC590DP |
C |
Si |
Mn |
P |
S |
Fe |
|
0.07 |
0.14 |
1.44 |
0.013 |
0.002 |
98.335 |
|
(2) The presentation of the images is well.
We thank the reviewer for the encouraging comment. We hope to present many new findings in the near future.
(3) The interfacial metallurgical bonding and microstructure evolution of the joints should be addressed.
We have not performed metallurgical bonding and microstructure evolution of the joints because we do not expect much change in the structure of the underlying material. The SPR process is a pure plastic deformation process that does not involve any external heat source (thus no cooling). Therefore, only grain boundary elongation is expected due to the SPR process at the limited area where plastic deformation takes place (we think that this part is of little interest to the readers of the journal related to manufacturing).
The following improvements in page 5 were made in the manuscript:
“Figure 5 shows the cross-section of the joints formed using the various types of die geometries shown in Figure 2 for two different panel configurations, namely, vibration-damping steel (top)/SPFC590DP (bottom) and SPFC590DP (top)/vibration-damping steel (bottom). Good interlock between the two sheets is ob-served for all die types and panel configurations. The figure shows that when the vibration-damping steel panel is on top, there exists some separation between the vibration-damping panel sheets within the rivet. The figure also shows that when the SPFC590DP panel is on top, the cavity formed within the rivet after joining can be significantly large for flat die types. Such separations and cavities can be mitigated using different die geometries such as the nipple type. Nevertheless, the subsequent tensile shear test reveals that any separations and cavities do not affect the overall joint quality. The cross-section images reveal that there are minimal microstructural changes in both the rivet and the base material. This is expected since no external heat is applied to the SPR process and the extent of plastic deformation is below the fracture point of the base material. Any microstructural change should be avoided in the SPR process since inhomogeneous material properties are not desirable to ensure joint quality consistency.”
(4) What is the shear stability of the joint? How many shear tests were done? This can be explained in the Materials and Methods.
In the materials and methods sections, we have included that the shear tests were done 3 times and the values presented here are the average of the three values. “Crosshead speed of 5 mm/min was used and the average value of three samples are presented here.”.

Round 2
Reviewer 1 Report
Thank you for addressing my previous remarks. Regarding the first one, the authors only referred to the RSW method for dissimilar joining. Since the aim is the automotive industry, typical processes such as laser welding (and its variants), which can be used to weld dissimilar materials, should be included in this discussion.
Author Response
The response and the changes are attached.

Reviewer 2 Report
The draft can be accepted in its current form.
Author Response
Thank you!